# Functional analysis of N-acetylglucosaminyltransferase-I knockdown in 2D and 3D neuroblastoma cell cultures

**M. Kristen Hall, Adam P. Burch**[ID]**, Ruth A. Schwalbe**[ID]*

Department of Biochemistry and Molecular Biology, East Carolina University, Greenville, North Carolina, United States of America

* schwalber@ecu.edu

**Data Availability Statement:** All relevant data are within the manuscript and its supporting information files.

## Abstract

Tumor development can be promoted/suppressed by certain N-glycans attached to proteins at the cell surface. Here we examined aberrant neuronal properties in 2D and 3D rat neuroblastoma (NB) cell cultures with different N-glycan populations. Lectin binding studies revealed that the engineered N-glycosylation mutant cell line, NB_1(-*Mgat1*), expressed solely oligomannose N-glycans, and verified that the parental cell line, NB_1, and a previous engineered N-glycosylation mutant, NB_1(-*Mgat2*), expressed significant levels of higher order N-glycans, complex and hybrid N-glycans, respectively. NB_1 grew faster than mutant cell lines in monolayer and spheroid cell cultures. A 2-fold difference in growth between NB_1 and mutants occurred much sooner in 2D cultures relative to that observed in 3D cultures. Neurites and spheroid cell sizes were reduced in mutant NB cells of 2D and 3D cultures, respectively. Cell invasiveness was highest in 2D cultures of NB_1 cells compared to that of NB_1(-*Mgat1*). In contrast, NB_1 spheroid cells were much less invasive relative to NB_1(-*Mgat1*) spheroid cells while they were more invasive than NB_1(-*Mgat2*). Gelatinase activities supported the ranking of cell invasiveness in various cell lines. Both palladin and HK2 were more abundant in 3D than 2D cultures. Levels of palladin, vimentin and EGFR were modified in a different manner under 2D and 3D cultures. Thus, our results support variations in the N-glycosylation pathway and in cell culturing to more resemble *in vivo* tumor environments can impact the aberrant cellular properties, particularly cell invasiveness, of NB.

## Introduction

Neuroblastoma (NB), a tumor of neuronal origin, is most frequently found in infants and young children and is often associated with a bleak prognosis [1]. NB exists as one of the most common solid extracranial tumors in children, resulting in nearly 15% of pediatric cancer-related deaths [2]. While some NBs proliferate slowly and ultimately spontaneously regress, others proliferate and spread quickly [1]. These more aggressive forms of NB have proven challenging to treat, with only 40% of high-risk patients achieving long-term survival, despite

**Funding:** This work was supported in part by National Institute of Health Grant GM129679 (to R. A.S.). The funders had no role in study design, data collection and analysis, decision to publish, or preparation of the manuscript.

**Competing interests:** The authors have declared that no competing interest exist.

recent therapeutics somewhat improving outcomes [1, 3]. Additional therapeutic approaches and treatments for NB are essential.

Glycosylation persists as the most complex post translational modification of proteins [1] and is implicated in numerous biological processes [4]. There are three types of *N*-glycans: oligomannose, hybrid, and complex. *N*-Acetylglucosaminyltransferases (GnTs), encoded by *MGAT* genes, create branch points on the conserved pentasaccharide of *N*-glycans to generate various types of *N*-glycans [4]. GnT-I initiates the conversion of oligomannose type *N*-glycans to hybrid type, which is further processed to complex type via GnT-II. Although it is known that alterations in glycan structure or glycan-glycan interactions effect cell proliferation and invasion [1, 2], uncertainty remains as to the roles of specific glycans on cancer development and progression [1]. For this reason, evaluation of N-glycans in cancer and glycan-based therapies are an attractive area of research for therapeutic development for cancer treatment.

Abnormal cell proliferation is essential for the progression of cancer and formation of tumors, of which, are capable of invading surrounding tissue. Traditionally, two-dimensional (2D) cell culture has been the foundation to investigate NB. Albeit these studies provide some insight into the development and progression of NB, 2D cell culture studies are insufficient in representing the three dimensional (3D) *in vivo* environment [5]. Spheroid cells, a 3D cell culture, more closely mimic the heterogeneity of *in vivo* tissues, including tumors, such that spheroids contain different populations of proliferating, differentiating, and viable cells in their various layers [6]. Studies of 2D and 3D cell cultures have demonstrated variation in expression of a tumor-associated antigen in carcinoma [7], protein expression patterns in human NB cell lines [5], and cancerous cellular processes [8]. The impact of 3D cultures on NB cell properties would contribute to a more comprehensive understanding of factors that promote and suppress NB.

Overexpression of matrix metalloproteinases (MMP), paladin, and vimentin are linked to modified cell morphology and invasiveness in cancer cells [9–11]. MMPs increase the availability of growth factors and cytokines to receptor tyrosine kinases (RTKs) by degrading the extracellular matrix [12]. Palladin, an actin cytoskeleton protein, is vital in neuronal maturation and development as well as neurite outgrowth [13]. Changes in cell motility and invasiveness have also been linked to alterations in the expression of the intermediate filament protein, vimentin [11, 14]. The RTK, epidermal growth factor receptor (EGFR), and glycolytic protein hexokinase-2 (HK2) influence cell growth and the overexpression of both have been linked to more invasive and aggressive cancers [15–17]. Overexpression of HK2 has been associated with as much as 70% of cancers [15] and is a known participant in the Warburg Effect. We postulate that the expression and activity of these and other candidate proteins are dependent on the types N-glycans populating NB cells.

Previously, our lab demonstrated that knockdown of *MGAT2/Mgat2* gene suppresses tumorgenicity in 2D cultured human [18] and rat NB cell lines [19, 20]. The knockdown of GnT-II prevents the conversion of hybrid to complex type N-glycans, thus yielding a glycosylation mutant cell line. Further, substitution of complex with hybrid types of N-glycans diminished aberrant NB cell properties [18, 20]. Given the implications of alterations in N-glycan processing on suppression/promotion of cancerous cell properties, herein, we further manipulate the N-glycan pathway via knockdown of *Mgat1*, thereby creating a mutant cell line with even less processed N-glycans. Lectin binding studies were employed to show that the N-glycans were mainly oligomannose type. In addition to our aim of examining changes in N-glycan processing on cell growth, dissociation, migration, and invasion, we aim to determine the impact of the microenvironment in which the cells are cultured, 2D versus 3D on those same cellular properties. Also, western blotting was used to distinguish if EGFR, hexokinase 2, palladin, and vimentin protein levels were altered among the three distinct cell lines, and further

compare the levels between 2D and 3D cultured cell lines. Gelatin zymography assays were utilized to detect differences in MMP activity. Thus, this report demonstrates that interruption in the N-glycosylation pathway alter aberrant NB cell properties, and furthermore that 2D and 3D cultures are affected in a different manner.

## Materials and methods

### Cell lines, cell culture, and cell transfection

Rat B35 neuroblastoma (NB) (American Type Culture Collection, Manassas, VA, USA) were used to generate a clonal NB cell line, NB_1 [19]. Previously, we used CRISPR/Cas9 technology to silence rat *Mgat2* gene [19] and those protocols were utilized to silence rat *Mgat1* in the NB_1 clonal cell line, generating N-glycosylation mutant cell lines, referred to as NB_1 (-*Mgat2*), and NB_1(-*Mgat1*), respectively. In brief, Zi-Fit Targeter software was used for designing *Mgat1* sgRNA oligonucleotides 5'−**CACC**GTGCTATCATCTTTGTGGGC−3' and 5'−**AAAC** GCCCACAAAGATGATAGCAC. Recombinant pSpCas9(BB)-2A-Puro vector (Addgene plasmid ID: 48139) encoding *Mgat1* was employed to create the GnT-I knockout in NB_1 cells, referred to as NB_1(-*Mgat1*). After transfections, genomic DNA was isolated from some cells of a colony of NB_1(-*Mgat1*) cells and a fragment of the *Mgat1* gene was amplified and cloned into pCR2.1 TOPO vector for DNA sequencing. DNA sequencing of 9 recombinant vectors showed frameshift mutations in *Mgat1* and was used as the NB_1(-*Mgat1*). Further there was no alternative start codon, in frame, until residue 251. The remaining portion of cells were expanded and used for subsequent experiments. DMEM containing 10% FBS, 50 U/ mL penicillin, and 50 μg/mL streptomycin at 37˚ was used in a 5% $CO_2$ atmosphere to maintain 2D and 3D NB cell cultures. The NB_1(-*Mgat1*) cell line was rescued by transient transfection with pCDNA3.1 recombinant vector coding the mouse *Mgat1* cDNA using the Lipofectamine® 2000 (Thermofisher Scientific, MA, USA) protocol [18]. The mouse *Mgat1* coding sequence was obtained from Dr. Pamela Stanly, College of Albert Einstein [21] and was modified to have BamHI at both 5' and 3' ends using PCR and then cloned into pCDNA3.1 vector at the BamHI restriction site for expression in mammalian cell lines.

### Whole cell lysates, total membranes and conditioned serum-free medium

RIPA buffer (PBS, 1% Triton X-100, 0.5% sodium deoxycholate, 0.1% SDS) plus protease inhibitor cocktail set III (EMD Biosciences, San Diega, CA, USA) was used to make whole cell lysates from 2D and 3D cell cultures, as we previously described [18]. Total membranes of 2D cell cultures, along with 3D cell cultures were purified as described [19]. Conditioned serum-free media was prepared as described [18], except cells were cultured in serum-free medium for 72–80 h, instead of 48 h. Further both 2D and 3D cell cultures were used to generate conditioned medium. In all cases, protein levels were determined by a modified Lowry assay. Whole cell lysates, total membrane, and conditioned medium samples were denatured and reduced in SDS-PAGE sample buffer, containing DTT for Coomassie blue stained gels and western blotting. Concentrated conditioned serum-free media samples were denatured in SDS-PAGE sample buffer for gel zymography. In all cases, aliquots of samples were stored at −20 or −80˚C for future use.

### 2D cell proliferation assay

Cell proliferation was measured by employment of 5-bromo-2-deoxyuridine (BrdU) proliferation assays (Millipore, Billerica, MA) following the manufacturer's protocol as previously documented [18]. 2 X $10^4$ cells were plated in 96 well plates. Plated cells were incubated for 24 h

before being fixed for 30 min at room temperature. Fixed cells were incubated for 1 h with anti-BrdU monoclonal antibody before a 30 min incubation with secondary goat-anti mouse IgG peroxidase conjugate. Absorbance was read at 450 nm using a Multiskan FC plate reader (Fisher Scientific, Atlanta, GA, USA) following a 30 min incubation with peroxidase substrate.

## 2D cell morphology assay

Cells were plated at low density on poly-lysine coated culture dishes (Mat-Tek, Ashland, MA, USA) and were incubated for 18 h before being imaged using a 10X objective on an Olympus IX71 microscope. Using Image J Software, outgrowth length and width were measured from the beginning of the cell body and halfway along the outgrowth length, respectively.

## 2D cell dissociation assay

Confluent CellBind Culture Dishes (Corning, NY, USA) were rinsed 2 times with DMEM before adding serum free media and detaching the cells using a cell scraper. Detached cells were pipetted ten times with a 1 mL pipet tip. Using a 10 X objective from a IX71 Olympus microscope, 25–30 fields per dish were captured for analysis of cell aggregates using Image J Software.

## Anchorage independent growth assay

To measure anchorage independent growth, the soft agar assay was utilized. The assay consists of two layers: the bottom layer consisting of 1% low melting temperature agarose in a 1:1 ratio with DMEM with 10% FBS and the bottom layer being 0.6% noble agar mixed 1:1 with the cell solution. The top layer is added only after the bottom layer has fully solidified. 6 well plates were used and were cultured for 14 days with extra DMEM media being added every 2 days to ensure cell viability and to prevent the agar from drying. A 4 X objective from an Olympus IX73 microscope was used to image the assay. Image J Software was used to quantify the area and number of the colonies.

## Cell migration and cell invasion assays

BD Falcon cell chambers (BD biosciences, CA, USA) were employed to evaluate cell migration. $2.5 \times 10^4$ cells in 500 μL serum free DMEM were added to each transwell insert without matrigel. 500 μL NIH-3T3 conditioned media was added to the lower chambers and cells were cultured for 20 h at 37˚. BD Falcon Matrigel invasion chambers (BD biosciences, CA, USA) were used to measure cell invasion following the manufacturer's instruction and with modifications as described [18]. For both assays, samples were done in quadruplet for 2 experiments. Membranes without and with Matrigel were removed from inserts, and then washed, fixed with 100% methanol, and stained with 1% Toluidine blue. Each membrane containing cells were counted using a Nikon TMS microscope. An Olympus IX73 was used to image the cells, to determine the number of either migratory or invasive cells per well.

**3D Spheroid cell formation, growth and morphology.** Uniform spheroids were used for 3D cell culture of the various NB cell lines. Medium (0.5 mL) was added to the 3D wells of Sphericalplate 5D plates (Kugelmeiers, Erlenbach, Sweden), followed by a 3 min 500 x g spin to eliminate potential air bubbles in the 3D wells. Cells ($7.5 \times 10^4$ cells/mL) were added to each prepared 3D well, and subsequently the plate was lightly tapped to evenly distribute the cells. DMEM medium was added every 2–3 days to ensure cell viability and health. All spheroids were cultured in this manner to produce consistent spheroid cells. Images were taken at days 1, 4, 6, 7, and 9 using an Olympus IX71 microscope's 10X objective to allow for the

quantification of the size of the spheroid cells with Image J Software. Spheroid cells cultured for 7 days were also imaged with a 40X objective to measure the diameters of peripheral cells to compare cell morphology. Peripheral Cells that were overlapping another cell, or in cases where there was no clear border to define the individual cell, were excluded from the analysis.

## Spheroid cell invasion assay

Cells were cultured for 1, 4 and 6 days in Sphericalplate 5D plates to form various size spheroids. Spheroid cells were collected from the 5D plates by placing them in a 15 mL centrifuge tube and allowed to settle for 10 minutes at room temperature. The suspended spheroid cells (40 μL) were then combined with Matrigel (200 μL). Spheroid/Matrigel mixture (40 μL) was pipetted into the center of the wells of a 24-well plate (4 wells per cell line) via pre-chilled pipet tips. Matrigel was polymerized by incubation at 37˚ for 30 minutes and then media (1 mL) was added to culture cells for 16 h or 24 h. Acquisition and analysis was similar to that previously described [22]. A 4X objective on an Olympus IX73 microscope was used to capture images. ImageJ software was utilized to measure the sphere and invasive areas. Cell invasion is reported as the ratio of the invasion area to the sphere area.

## Coomassie blue stained gels and western blotting

Total cell membrane, concentrated conditioned serum-free media, and whole cell lysate samples were evaluated by coomassie blue stained electrophoreses gels and western blotting [18]. Proteins were electrophoresed on 10% SDS gels and stained with coomassie blue to evaluate total protein levels. For Western blotting, PVDF membranes (Millipore, Billercia, MA, USA) containing separated proteins on 10% SDS gels were incubated with primary and secondary antibodies and then visualized using NBT/BCIP. The primary antibodies include the following: rabbit anti-palladin antibody (Proteintech, Rosemont, IL), rabbit anti-EGFR (Cell Signaling Tech, Danvers, MA), rabbit anti-hexokinse 2 (Abcam, Cambridge, MA), and rabbit anti-vimentin (Cell Signaling Tech, Danvers, MA).

## Gelatin zymography assay

Proteins of non-reduced conditioned serum-free media samples were separated on 10% SDS acrylamide gels, plus 4 mg/mL gelatin at 20 mA for 1.5 h. Gels with electrophoresed proteins were washed twice with washing buffer (2.5% Triton X-100, 50 mM Tris-HCL, pH 7.5, 5 mM $CaCl_2$, 1 μM $ZnCl_2$) for 30 minutes, and then rinsed in incubation buffer (1% Triton X-100, 50 mM Tris-HCL, pH 7.5, 5 mM $CaCl_2$, 1 μM $ZnCl_2$) for 5–10 minutes with gentle agitation at 37˚C. The gel was then incubated in fresh buffer at 37˚C with slight agitation. After 24 h, the gel was stained with Coomassie Blue and bands were visualized by incubation of gel in a destaining solution.

## Data analysis

Image J software was employed for data analysis, including measurements of individual cells, cell aggregates, spheroid cells, and spheroid invasion. Adobe Photoshop was employed for microscopic images, gelatin gels and western blot images. Origin 9.55 was used for graphics and statistics. UN-SCAN-IT and image J softwares were employed to determine total intensity, and the relative size and darkness of a given band, respectively. Data is presented as the mean S.E. where n denotes the number of observations, as indicated. Statistical comparison of two groups were accomplished using student's t-test while one-way ANOVA with Bonferroni adjustments was used for more than two groups while the unpaired student's t-test was used for comparing mean values, unless indicated.

# Results

Using CRISPR/Cas9 technology, the *Mgat1* (GnT-I) gene was silenced in NB_1 cells, the parental cell line, and was named the NB_1(-*Mgat1*) cell line. The parental cell line, NB_1, has GnT-I and GnT-II, and therefore has all three types of N-glycans while the N-glycosylation mutant cell line, NB_1(-*Mgat1*) has only oligomannose N-glycans (Fig 1A). DNA sequencing of 9 recombinant vectors that encoded a fragment of the *Mgat1* gene showed that 6 *Mgat1* DNA fragments had a G residue inserted after the 54th nucleotide residue while 3 others had nucleotide 54 removed (Fig 1B). Both indels produced frameshift mutations with an alternative start site at amino acid residue 251. To examine whether the N-glycans were changed in the N-glycosylation mutant cell line, as expected from silencing the *Mgat1* gene, we compared lectin binding studies of the NB_1(-*Mgat1*) cell line to the parental (NB_1) cell line. Further we examined whether NB_1(-*Mgat1*) cells transiently expressing *Mgat1*, referred to as NB_1 (-/+*Mgat1*) cells, produced more processed N-glycan structures, such as hybrid and complex types. The rescue would support changes in N-glycan structures of the NB_1(-*Mgat1*) cell line

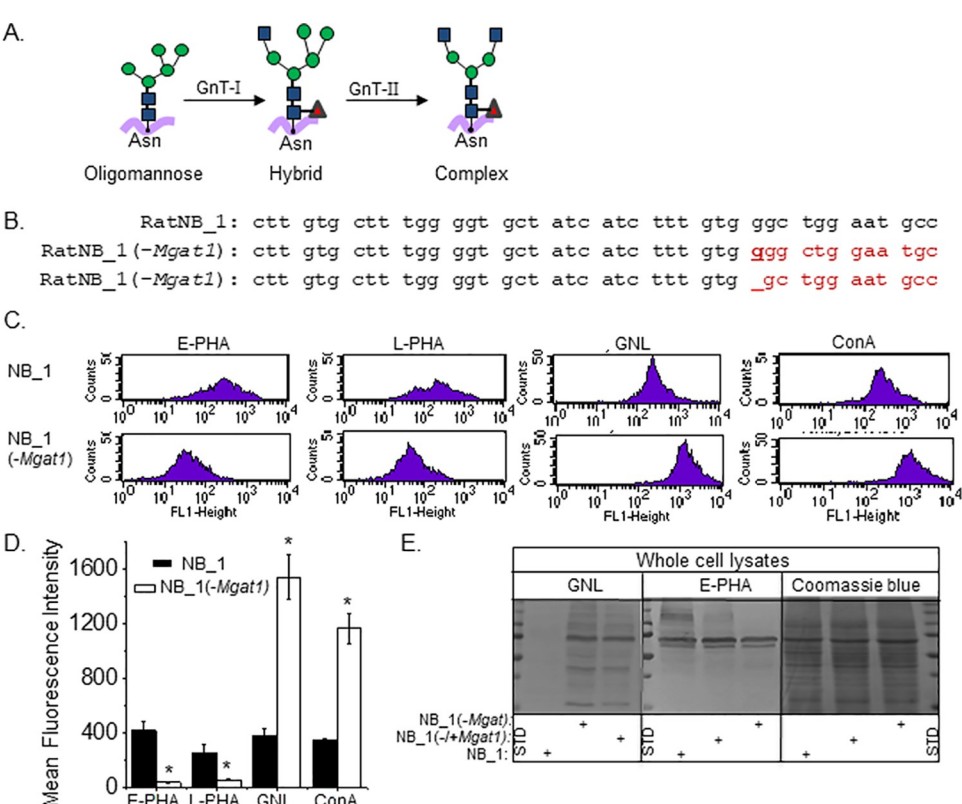

**Fig 1. Characterization of a neuroblastoma cell line with knockout of GnT-I.** (A) The simplest N-glycan is oligomannose which gives rise to more mature N-glycans, hybrid and complex. GnT-I converts oligomannose to hybrid, and subsequently GnT-II converts hybrid to complex. (B) The coding sequence of rat *Mgat1* from 25 to 66 was compared to that isolated from a newly created NB_1(-*Mgat1*) cell line. Two frameshift mutations were identified, including an inserted g nucleotide and a deletion of a g nucleotide as indicated in bold red font and dash line, respectively. Further the next in-frame start codon does not occur until residue 251 (C) Typical flow cytometry plots of NB_1 (top panels), and NB_1(-*Mgat1*) (bottom panels) cell lines interacting with fluorescently labelled lectins. (D) Mean fluorescence intensity of E-PHA (*n* = 5), L-PHA (*n* = 5), GNL (*n* = 5) and ConA (*n*≥2) bound to NB_1 and NB_1 (-*Mgat1*) cells. Graph denotes mean ± SEM and were compared by student's t-test (*p < 0.01). (E) Lectin blots and a coomassie blue stained gel of proteins from whole cell lysates of NB_1 and NB_1(-*Mgat1*) cell lines. Vertical dotted and solid lines denote gel lane dividers and a different gel, respectively. Molecular weight standards (STD) in kDa: 250; 150; 100; 75; 50; and 37 from top to bottom.

was due to silencing of the *Mgat1* gene, and not an off-target gene. Fluorescent-labelled lectins were allowed to interact with live NB_1(-*Mgat1*) or NB_1 cells, and then fluorescence intensity in various cell samples were measured by flow cytometry as viewed in the representative histograms (Fig 1C), and mean fluorescence intensity of each of lectins for both cell lines determined (Fig 1D). Markedly greater amounts of *Phaseolus vulgaris* Erythoagglutinin (E-PHA), a lectin with very high affinity for complex type N-glycans with bisecting N-acetylglucosamine (GlcNAc) relative to other N-glycan structures [23], bound to parental cells than N-glycosylation mutant cells. *Phaseolus vulgaris* Leucoagglutinin (L-PHA), which has much higher affinity for complex type N-glycans than hybrid and oligomannose types [23], bound to parental cells at a higher level than that of N-glycosylation mutant cells. More *Galanthus nivalis* Lectin (GNL) and concanavalin A (ConA) bound to the N-glycosylation mutant cells than parental cells. Both lectins have higher affinities to oligomannose type of N-glycans than other types [23, 24]. Predominant types of N-glycans identified by flow cytometry were verified by lectin blotting of whole cell lysates from NB_1 and NB_1(-*Mgat1*) cell lines (Fig 1E). The GNL interaction was detected much more strongly with proteins from NB_1(-*Mgat1*) relative to those from NB_1. On the other hand, the E-PHA interaction was observed more strongly with proteins from the NB_1 cell line than those from the NB_1(-*Mgat1*) cell line. Further the detected interaction of GNL and E-PHA with separated proteins from NB_1(-*Mgat1*) cells transfected with *Mgat1* was diminished and improved, respectively, compared to those from the NB_1 (-*Mgat1*) cell line. The Coomassie blue stained gel confirmed the levels of protein loaded per well for the lectin blots were quite similar in various cell lines. Overall, the lectin binding studies confirmed that the NB_1 cell line expressed the mature N-glycans type, complex, and had significant levels of bisecting type N-glycans [19, 20] while the NB_1(-*Mgat1)* cell line had oligomannose type. Further when NB_1(-*Mgat1*) cells were transfected with *Mgat1* they were observed to produce higher branched N-glycans, indicating that the changes in the N-glycan structures were due to silencing *Mgat1* in the glycosylation mutant cell line.

## Morphology of individual and spheroid NB cells with modifications in the N-glycosylation pathway

To examine whether changing N-glycan types alters NB cell morphology, the NB_1 and NB_1 (-*Mgat1*) cell lines were employed to evaluate the impact of complex and oligomannose types of the N-glycans. Previously, we showed that neurites were shortened when comparing NB cells expressing hybrid versus complex types of N-glycans [20]. Images of individual cells were captured (Fig 2A) and the length (Fig 2B) and width (Fig 2C) of neurites were measured in NB_1 and NB_1(-*Mgat1*) cells. Images revealed rounder cells with shortened neurites of the mutant cell line. The length of the neurites was reduced by about 70% while the width of the neurites was increased by 1.3-fold for NB_1(-*Mgat1*) relative to NB_1 cells. Morphological studies of spheroid cells were used to ape the environment of cells in a multicellular organism. Micrographs were obtained from spheroid cells cultured for 7 days from NB_1, NB_1(-*Mgat2*), and NB_1(-*Mgat1*) cell lines (Fig 2D). The NB_1(-*Mgat2*) cell line was employed to address morphological changes of spheroid NB cells expressing significant levels of hybrid type N-glycans [19, 20]. Parental cells formed larger spheroids than the N-glycosylation mutant cells lines (Fig 2E). Further the mutant cell line that could synthesize oligomannose and hybrid types of N-glycans formed larger spheroid cells than those that could solely synthesize oligomannose type of N-glycans. Images also revealed that cells at the edge of the spheroids, as denoted by the blue arrows, were larger for cell lines that formed larger spheroids (Fig 2F). Thus, neurite formation in the 2D cell cultures and spheroid cell formation of the 3D cultures were lessened when the N-glycosylation pathway was prevented from synthesizing higher ordered N-glycans.

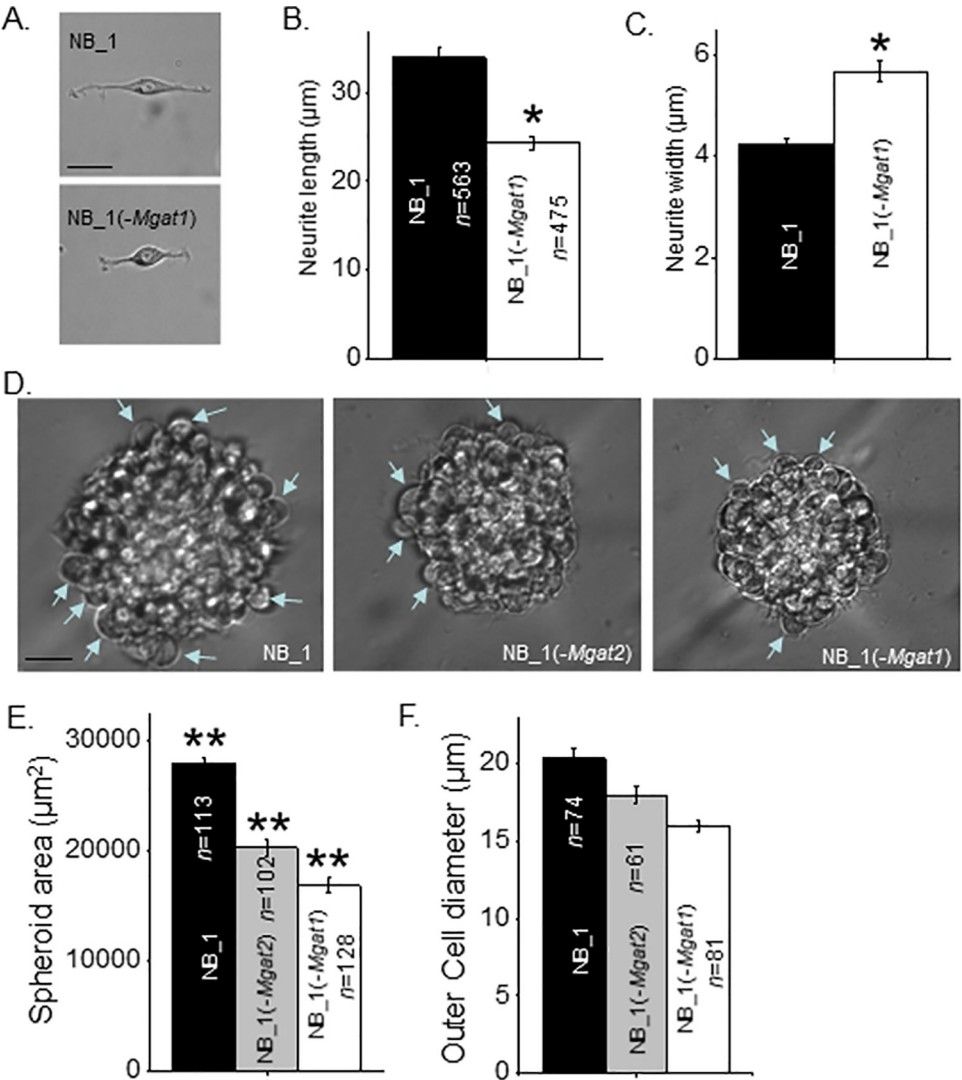

**Fig 2. Morphology of NB cells and NB spheroids are dependent on the type of N-glycan.** (A) Representative images of NB_1 and NB_(-*Mgat1*) cells from 2D cell cultures. Length (B) and width (C) of neurites from cells grown in 2D cell cultures. (D) Typical images of 7 days old spheroids obtained at 40x magnification. Blue arrowhead depicts measured cells at outer edge of sphere. (E) Quantification of spheroid size determined from images obtained at 10x magnification. (F) Diameter of cells at outer edge of sphere were quantified. Scale bars are 25 μm. Data are presented as the mean±SEM and were compared by one-way ANOVA followed by Holm-Bonferroni adjustment (*p $< 0.02$, **p$<0.001$). *n* denotes the number of neurites, cell diameters or spheroids measured, respectively.

## Cell proliferation and growth of 2D and 3D cell cultures with interruptions in the N-glycosylation pathway

Cell proliferation was measured in 2D sub-confluent cell cultures by incorporation of BrdU into the DNA during its replication process (Fig 3A). There was about a 54% decrease in cell proliferation between NB_1(-*Mgat1*) cells and NB_1 cells. When NB_1(-*Mgat1*) cells were transient transfected with *Mgat1* cDNA, cell proliferation was increased to near 80%. The rescued cells, NB_1(-/+*Mgat1*), indicated that the slowed cell proliferation in NB_1(-*Mgat1*) cells was due to silencing the *Mgat1* gene. As such, higher ordered N-glycans enhanced the cell proliferation rate of NB cells. Cell growth was monitored under 3D cell cultured conditions.

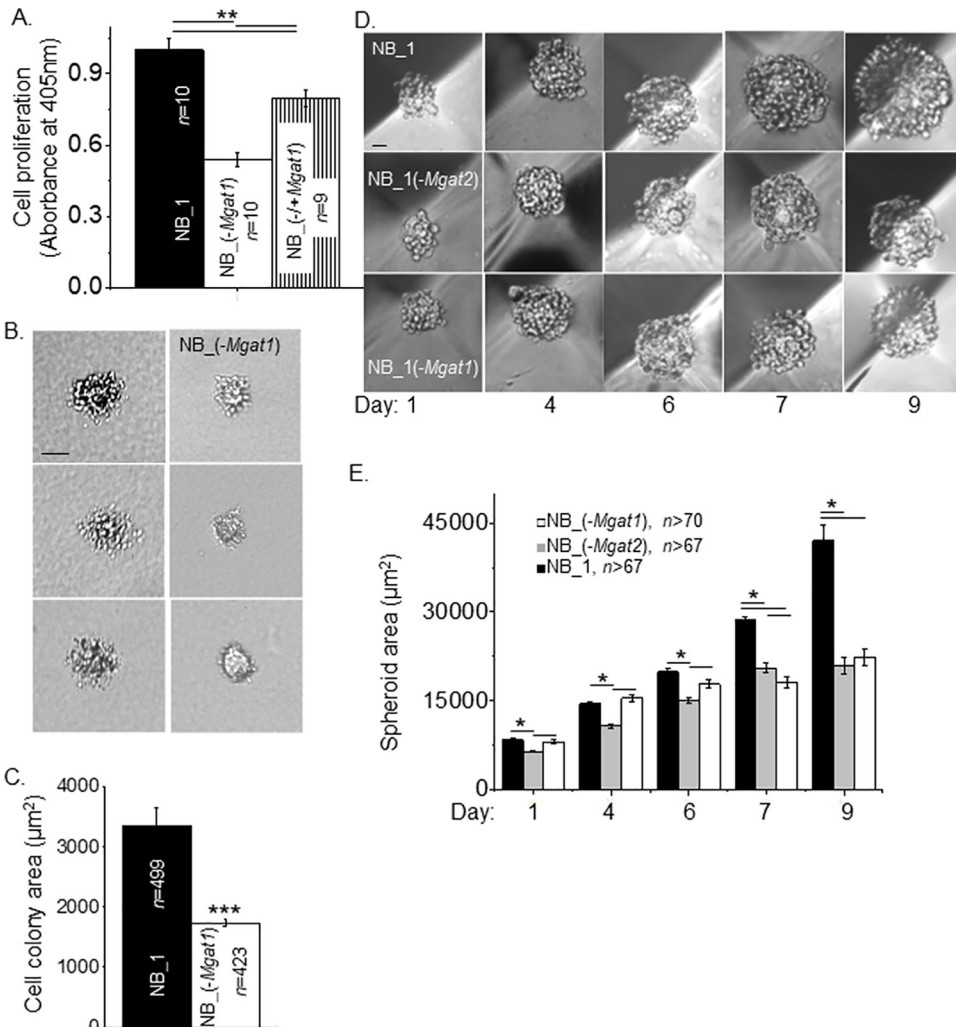

**Fig 3. Diminution of N-glycan diversity alters NB cell growth and spheroid formation.** (A) Cell proliferation of NB_1 and NB_(-*Mgat1*) cell lines from 2D cell cultures, and the rescued cell line, NB_(-/+*Mgat1*). (B) Images of growing cell colonies obtained from anchorage independent cell growth of NB_1 and NB_(-*Mgat1*) cell lines. (C) Quantification of the size of cell colonies after 13 days of growth. (D) Representative images of spheroid formation in NB cell lines at various time points were obtained at 10x magnification. (E) Quantification of spheroid growth from 1 to 9 days in culture. Pearson's correlation coefficient of the spheroid area as a function of days for NB_1, NB_1 (-*Mgat2*) and NB_(-*Mgat1*) were 0.99, 0.99, and 0.94, respectively. Growth rates were (in area/days): 10,130, NB_1; 6,625, NB_1(-*Mgat2*); and 6,036, NB_1(-*Mgat1*). Scale bars are 25 μm. Data are presented as the mean±SEM and were compared by one-way ANOVA followed by Bonholm adjustment for more than 2 groups and student t-test for 2 groups, (*p < 0.05, **p<0.005, ***p<1 x10$^{-6}$). *n* signifies number of wells (*A*), cell clusters (*C*), and spheroids (*E*).

Images of cell colony size under anchorage independent cell growth conditions for 13 days were reduced in the glycosylation mutant cell line compared to the parental cell line (Fig 3B). Quantification of cell colony area between NB_1 and NB_1(-*Mgat1*) cell lines revealed close to a 52% difference in cell growth (Fig 3C). Lastly, cell growth was observed at 1, 4, 6, 7 and 9 days for NB_1, NB_1(-*Mgat1*) and NB_1(-*Mgat2*) cell lines using 5D spherical plates (Fig 3D). Spheroid cell size of NB_1 and NB_1(-*Mgat1*) cell lines were similar up to 6 days while the spheroids of NB_1(-*Mgat2*) cells were slightly smaller (Fig 3E). Though the spheroid size of NB_1(-*Mgat2*) cells was larger than that of NB_1(-*Mgat1*) cells at day 7 and also the spheroid from NB_1 cells continued to be the largest. The growth rates were (in spheroid area/days):

10,130, NB_1; 6,625, NB_1(*Mgat2*); and 6,036, NB_1(*Mgat1*). However, it took close to 9 days of spheroid growth for NB_1 cells to establish a 2-fold difference in spheroid size relative to either of the glycosylation mutant cells. The growth of the spheroid cells from 1 to 9 days displayed high linearity as Pearson's correlation coefficients were the following: NB_1, 0.95; NB_1(-*Mgat2*), 0.97; and NB_(-*Mgat1*), 0.98. Taken together, lessened cell growth caused by preventing the synthesis of complex and hybrid types of N-glycans in 2D and 3D cell cultures was a common trend. However, it should be noted that cell growth was hampered to a greater extent in 3D cell cultures than 2D cell cultures.

## Cell migratory rates were relatively constant for NB cell lines with different types of N-glycans

Cell chambers were divided into upper and lower chambers using membrane inserts. A low density of cells from monolayer suspensions were placed on top of membrane inserts to produce a 2D sub-confluent cell culture. Conditioned medium from NIH3T3 cell cultures was placed in bottom chambers to attract cells to migrate from top to bottom chambers. After a 20 h incubation, membrane inserts were collected, washed, and mounted on microscope slides, and then images of the membranes bottom-side were acquired. Micrographs showed that similar numbers of NB_1, and NB_1(-*Mgat1*) cells migrated to the bottom chamber (Fig 4A). Further cell migration was unaltered in the later cell line with transient expression of GnT-I. Quantification of the number of migratory cells from each cell line demonstrate the reproducibility of our observations (Fig 4B). These results indicate that the migratory rates in NB cells was unaltered by disrupting the N-glycosylation pathway.

## Cell invasiveness was lessened by decreased levels of complex type N-glycans in 2D NB cell cultures

Since cell migratory rates were unchanged in the various NB cell lines, similar cell chambers were used to directly assess cell invasiveness, except membrane inserts were layered with an extracellular matrix (Matrigel®). Again, addition of conditioned medium from NIH3T3 cell cultures to the bottom chamber was used to encourage cells to migrate to the bottom cell chamber but to get there they had to digest the matrigel. Representative images showed that there were more NB_1 cells than NB_1(-*Mgat1*) cells on the bottom-side of the membrane insert, and that expression of GnT-I in the latter cell line called NB(-/+*Mgat1*) increased invasive cell number (Fig 4C). The average number of invasive cells from each of the cell lines show that the NB_1 cell line was most invasive while the NB_1(-*Mgat1*) cell line was least invasive (Fig 4D). Cell invasiveness of NB_1(-*Mgat1*) was about 36% of NB_1, and when NB_1 (-*Mgat1*) cells were transfected with *Mgat1*, the cells were more invasive, such that NB_1 (-/+*Mgat1*) cells were about 76% of NB_1 cells. Thus, these results reveal that the decrease in cell invasiveness was due to preventing the conversion of oligomannose to complex types of N-glycans in 2D sub-confluent cell cultures.

## Cell invasiveness was intensified by lowered levels of complex type N-glycans in 3D NB cell cultures

Since a cancerous mass is a cluster of cells, we conducted cell invasion studies using spheroid cells. Spheroids were formed for 24 h in 5D spherical plates and added to cell dishes in a Matrigel slurry. After 24 h, images of the spheroids were obtained for NB_1, NB_1(-*Mgat2*) and NB_1(-*Mgat1*) cell lines (Fig 5A). Images revealed that NB_1(-*Mgat1*) cells had more and longer protrusions than the other types of cells, indicating that the glycosylation mutant cells

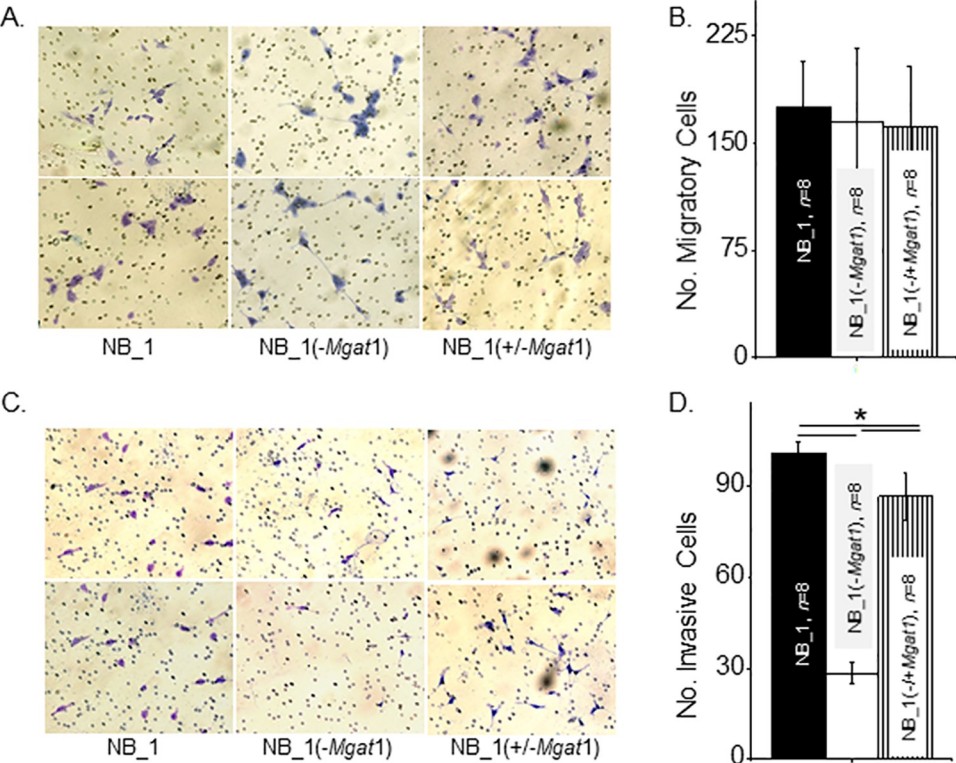

**Fig 4. Substitution of complex with oligomannose types of N-glycans on cell migration and invasion of sparse cells.** (A) Represented images of migratory parental and glycosylation mutant NB cells from the bottom of the membrane insert contained in transwell chambers after 20 h incubation, as indicated. (B) Quantification of the number of migratory cells that crossed the membrane insert per well. (C) Selected micrographs of invading parental and glycosylation mutant NB cells from Matrigel invasion chambers following a 20 h incubation period. (D) Average number of invasive cells per well. Bright purple migratory and invasive cells and pores in membrane are visible. Data are presented as the mean±SEM and were compared by one-way ANOVA using Holm-Bonferroni test (*p < 0.005). *n* denotes number of wells examined.

solely expressing oligomannose type N-glycans were more invasive than those expressing hybrid or complex types of N-glycans. To quantify cell invasiveness, the ratio of measured invasive area to the measured sphere area was determined as shown in Fig 5B. The cell invasiveness of the NB_1(-*Mgat1*) cell line was 2.3-fold greater than the NB_1 cell line and 3-fold greater than the NB_1(-*Mgat2*) cell line (Fig 5C). The Pearson's correlation coefficients were greater than 0.69 for each cell line (Fig 5D). In all cases, cell invasion was dependent on sphere area. Further, oligomannose type N-glycans promoted cell invasiveness relative to higher order N-glycans, such as hybrid and complex types.

Next, we addressed cell invasiveness due to different spheroid size and invasion incubation period of the various NB cell lines. Spheroid NB cells were grown for longer periods of time in 5D spherical plates and then cultured in a Matrigel slurry for 16 (panels A & B) or 24 hours (panel C) (Fig 6). The longer growth period allowed examination of cell invasiveness for larger spheroid cells of the various cell lines. It was observed that the NB_1(-*Mgat1*) cell line was more invasive at 4 (panel A) and 6 (panel B) days old spheroid cells than those of the NB_1 cell line. Further spheroid cells of NB_1(-*Mgat2*) were less invasive than the NB_1 cells. A similar trend was also observed when 6-day old spheroid cells invaded for 24 h (Fig 6C). Average cell invasion was about 2.1 for NB_1(-*Mgat1*) cells when spheroids cultured for 4 (Fig 6D) and 6 (Fig 6E) days invaded for 16 h while these values increased to close to 5.1 when 6 day old

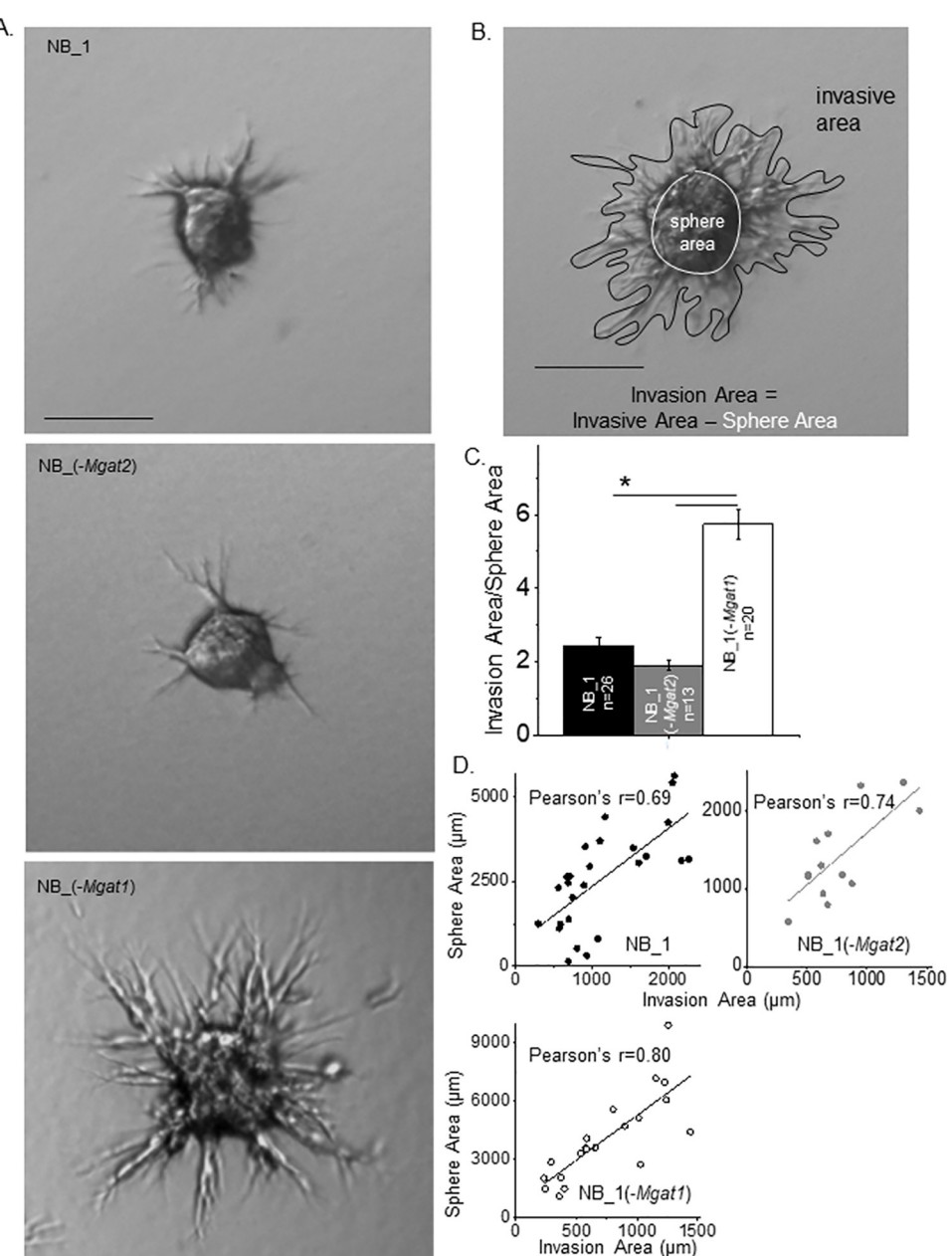

**Fig 5. Disruption in the N-glycosylation pathway modifies 3D cell invasion.** (A) Images of representative cell invasiveness from 1-day old spheroids of parental and N-glycosylation mutant cell lines cultured for 24 h in Matrigel matrix. (B) Micrograph of invading spheroid cells indicating measured sphere and invasive areas. Scale bars represent 25 μm. (C) The ratio of the invasion area to the sphere area were determined and plotted for NB_1 (*n* = 22), NB_1 (-*Mgat2*) (*n* = 15) and NB_1(-*Mgat1*) (*n* = 18) cell lines. (D) Pearson's correlation coefficients from plots of sphere area versus invasion area for each cell line. Data are presented as the mean±SEM and were compared by one-way ANOVA using Holm-Bonferroni test (*p < 0.05). *n* denotes number of spheroids measured.

spheroids invaded for 24 h, respectively (Fig 6F). Increases were also measured for average cell invasion values for NB_1 and NB_1(-*Mgat2*) when the invasion time was increased from 18 h to 24 h. When comparing all 3D spheroid invasion assays, cell invasiveness was greatest in the NB_1(-*Mgat1*) cell line, intermediate for the NB_1 cell line and least in the NB_1(-*Mgat2*) cell line. Based on Pearson's correlation coefficeints, cell invasion area was dependent on sphere

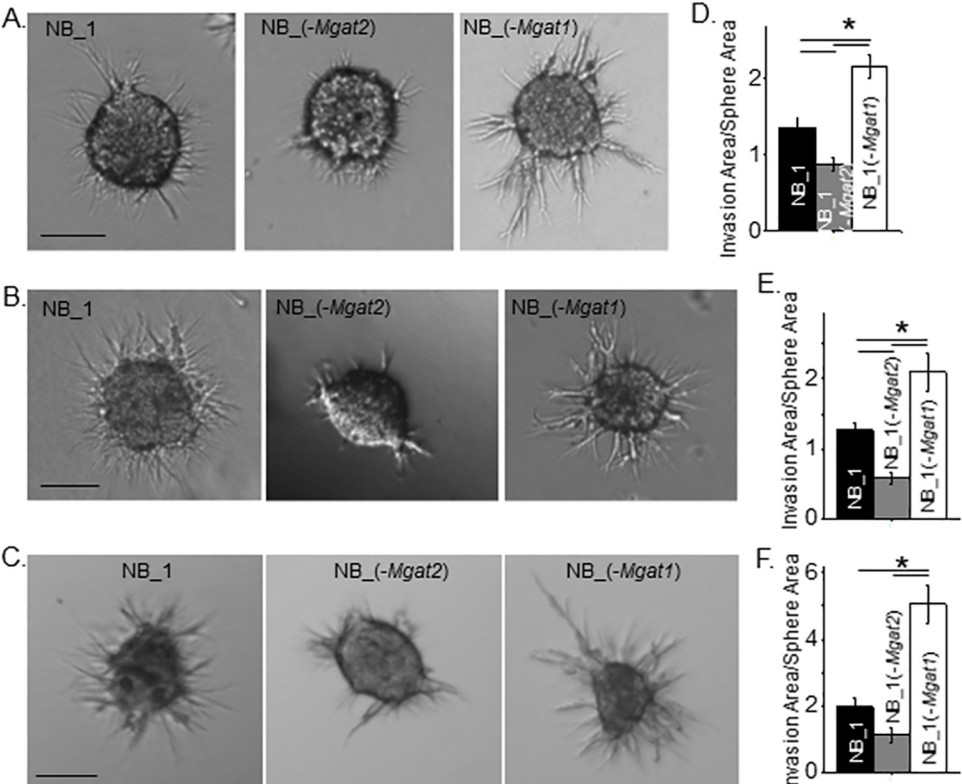

**Fig 6. NB cell invasion is independent of spheroid cell size and dependent on incubation period of spheroid cells.** Characteristic images of invading spheroid cells from those cultured for 4 (A) and 6 (B) days of NB_1 (*n* = 45 and 43), NB_1(-*Mgat2*) (*n* = 49 and 26) and NB_1(-*Mgat1*) (*n* = 59 and 48) cell lines and incubated in Matrigel matrix for 16 h duration, and spheroid cells cultured for 6 days (C) and incubated in Matrigel matrix for 24 h duration of NB_1 (*n* = 21), NB_1(-*Mgat2*) (*n* = 23) and NB_1(-*Mgat1*) (*n* = 20) cell lines. Scale bar denotes 25 μm. Quantification of cell invasiveness of the various cell lines in which spheroids were cultured for 4 (D) and 6 (E and F) days and incubated in Matrigel matrix for 16 (D and E) and 24 (F) hours. Data are shown as the mean±SEM and were compared by one-way ANOVA using Holm-Bonferroni post-hoc test (*p < 0.05). *n* denotes number of invading spheroids. Pearson's correlation coefficient determined from sphere area versus invasion area were the following: NB_1, 0.69; NB_1 (-*Mgat2*), 0.65; NB_1(-*Mgat1*), 0.75 from 4 day old spheroid allowed to invade for 16 h. Pearson's correlation coefficients for the following: NB_1, 0.57; NB_1(-*Mgat2*), 0.62; NB_1(-*Mgat1*), 0.51, and NB_1, 0.68; NB_1(-*Mgat2*), 0.51; NB_1(-*Mgat1*), 0.79 from 6 day old spheroid allowed to invade for 16 h and 24 h, respectively.

area for all three cell lines. Hence, cell invasiveness was modified by altering the N-glycosylation pathway in NB cells.

## MMP activities are altered by changes in the N-glycosylation pathway in both 2D and 3D cultures

Gelatin zymography of concentrated conditioned serum-free media of 2D cell cultures from NB_1, NB_1(-*Mgat1*) and NB_1(-*Mgat2*) cell lines revealed 3 intense bands of gelatinase activity at about 110 kDa (black arrow), 90 kDa (grey arrow), and 65 kDa (black arrow) (Fig 7A). The coomassie blue stained gel of identical loads indicate that similar amounts of protein were loaded. The upper band had greatest gelatinase activity in the NB_1(-*Mgat1*) cell line while the NB_1 and NB_1(-*Mgat2*) cell lines had comparable levels (Fig 7B). The middle band was most intense for NB_1, intermediate lightness for NB_1(-*Mgat2*) and least intense for NB_1 (-*Mgat1*). For the lower band, both NB_1 and NB_1(-*Mgat2*) had similar band intensity while that of NB_1(-*Mgat1*) was lower. It appears that NB_1 cells have higher gelatinase activity than

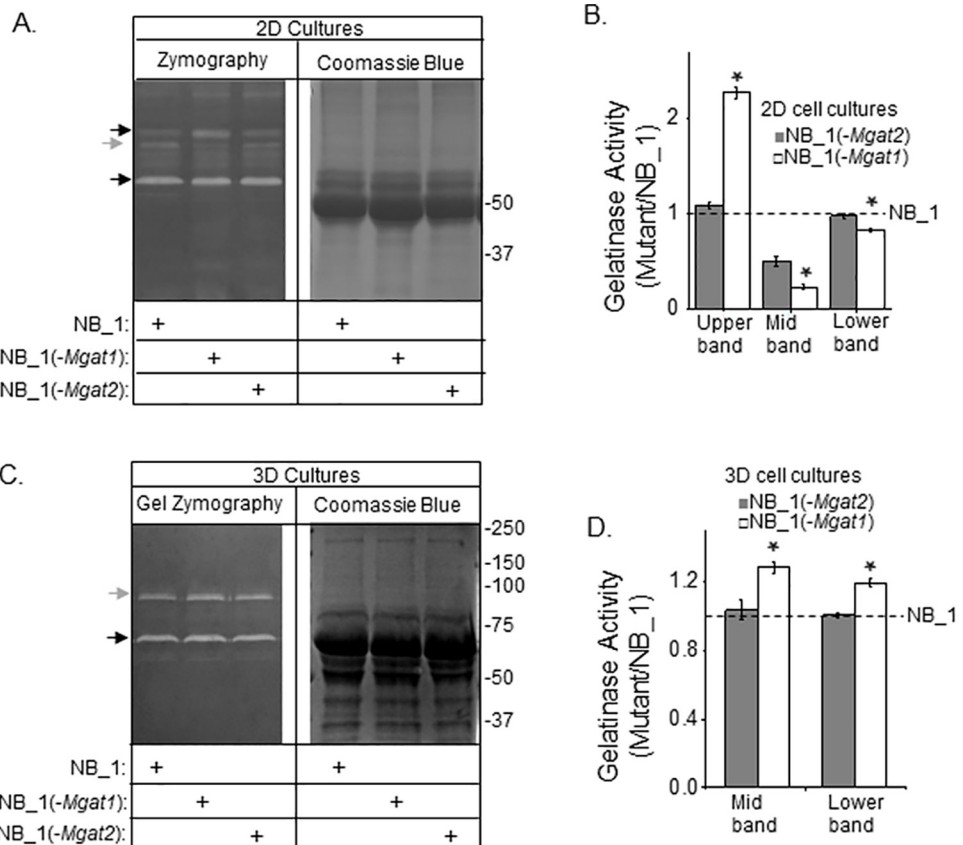

**Fig 7. Gelatinase activity of 2D and 3D NB cell cultures.** Gel zymography of conditioned serum-free medium samples from confluent plates (A) and spheroids cells (C) of NB_1, NB_1(-*Mgat1*) and NB_1(-*Mgat2*) cell lines (left panels). The amount loaded per well for 2D and 3D samples were 12 μg and 3 μg per well, respectively. Coomassie blue stained gels (right panels) of conditioned samples (10 μg) were loaded per well. Black arrows adjacent to gels denote upper and lower bands while grey arrow signifies middle (Mid) band. Numbers adjacent to gels denote molecular markers. Bands from glycosylation mutant samples were normalized to those NB_1 for 2D (B) and 3D (D) cell cultures. Data are shown as the mean±SEM and were compared by one-way ANOVA using Holm-Bonferroni post-hoc test (*p < 0.05). *n* = 3 for upper, middle and lower bands from the various samples.

the N-glycosylation mutant cell lines under 2D cell culturing conditions. Evaluation of the 3D samples by the gelatinase assay showed 2 intense bands of gelatinase activity at 90 kDa and 65 kDa (Fig 7C). Similar levels of proteins were loaded as shown in the coomassie blue stained gel. The lightness of both bands was highest for the NB_1(-*Mgat1*) sample while the lightness of the bands of the NB_1(-*Mgat2*) sample was quite like NB_1 (Fig 7D). Gelatinase activities were greater in the 3D cell cultures than 2D cultures as the amount of protein loaded for the 2D gel was close to 2-fold greater. Taken together, the results showed that the NB_1 cells have higher gelatinase activity than the N-glycosylation mutant cell lines under 2D cell culturing conditions while the gelatinase activity was highest for the NB_1(Mgat1) cells grown under 3D conditions.

## N-glycan types modify cell-cell adhesion strength in cellular monolayers and spheroid cells

The strength of cell-cell adhesion of NB_1(-*Mgat1*) cells was compared to that of NB_1 cells by breaking up detached cell monolayers. Representative images show that size of cell monolayer

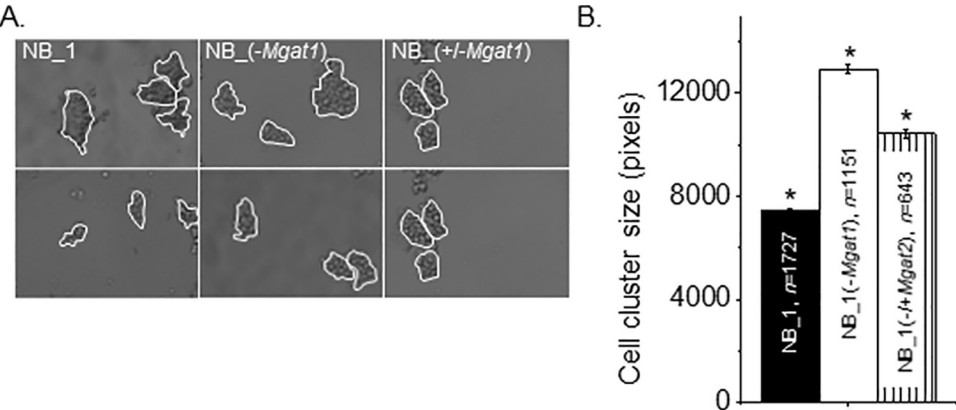

**Fig 8. Modification of expression levels of *Mgat1* impact cell-cell adhesion.** (A) Selected DIC images obtained from NB_1 (left panels), NB_1(-*Mgat1*) (middle panels), and NB_1(-/+*MGAT1*) (right panels) cell lines. Cell aggregates of more than 5 cells, as encircled in white, were analyzed. (B) The average area of cell aggregates from 3 cell dissociation experiments for each cell line. *$P<0.01$. Mean differences were compared using One-way ANOVA followed by Holmes-Bonferroni's test was used to compare differences in mean values. A value of $P<0.00000001$ was considered significant (*). *n* signifies number of cell clusters.

fragments from glycosylation mutant cells were larger than those from parental cells (Fig 8A). Further transfection of NB_1(-*Mgat1*) cells with the *Mgat1* gene reduced the size of the cell clusters. Mean cell cluster area was increased in NB cells by at least 1.7-fold when Gn-TI was knocked down and knock in of Gn-TI in some of the NB_1(-Mgat1) cells reduced the size to about 1.4-fold (Fig 8B). These results showed that lowered levels of complex and hybrid N-glycans strengthened the cell-cell interaction in monolayer NB cells.

## Palladin, vimentin and EGFR levels are changed by N-glycan types and 3D cultured cells

Western blotting was conducted to measure the protein abundance of HK2, palladin and vimentin in whole cell lysate samples of 2D and 3D cultures, along with EGFR in total membrane samples from both types of cultures (Fig 9A). Total protein levels were comparable for the whole cell lysate or total membrane samples from 2D or 3D cultures as shown in the coomassie blue stained gel (Fig 9B). Both palladin (NB1, 62±4%, *n* = 3; NB_1(-*Mgat2*), 73±10%, *n* = 3; and NB_1(-*Mgat1*), 59±8%, *n* = 3) and HK2 (NB1, 53±9%, *n* = 3; NB_1(-*Mgat2*), 48 ±10%, *n* = 3; and NB_1(-*Mgat1*), 49±5%, *n* = 3) protein levels were reduced in 2D cultures relative to those of 3D while vimentin and EGFR levels appeared similar between the culturing conditions. When *Mgat1* or *Mgat2* were silenced in the NB_1 cell line, the abundance of palladin was lowered, and significantly lowered in the NB_1(-*Mgat2*) compared to the NB_1 (-*Mgat1*) in 2D cell cultures (Fig 9C). The abundances of HK2 and EGFR proteins were unchanged by altering the glycosylation pathway in 2D cultures. Vimentin level of the NB_1 (-*Mgat2*) cell line was markedly lowered than that of the NB_1 cell line, and furthermore reduced to about 40% of the NB_1(-*Mgat1*) cell line in 2D cultures. In 3D cell cultures, the level of palladin was lowered when either *Mgat1* or *Mgat2* genes were silenced in NB_1, and palladin was also significantly lowered in the NB_1(-*Mgat1*) cell line compared to that of the NB_1(-*Mgat2*) cell line (Fig 9D). The level of HK2 was unaltered in 3D cultures. The abundance of vimentin in NB_1(-*Mgat1*) was significantly higher than that from NB_1(-*Mgat2*) while the later was quite like that of NB_1. EGFR levels were significantly lower for NB_1 (-*Mgat1*) relative to NB_1(-*Mgat2*), which was like that of NB_1. Thus, proteins involved in

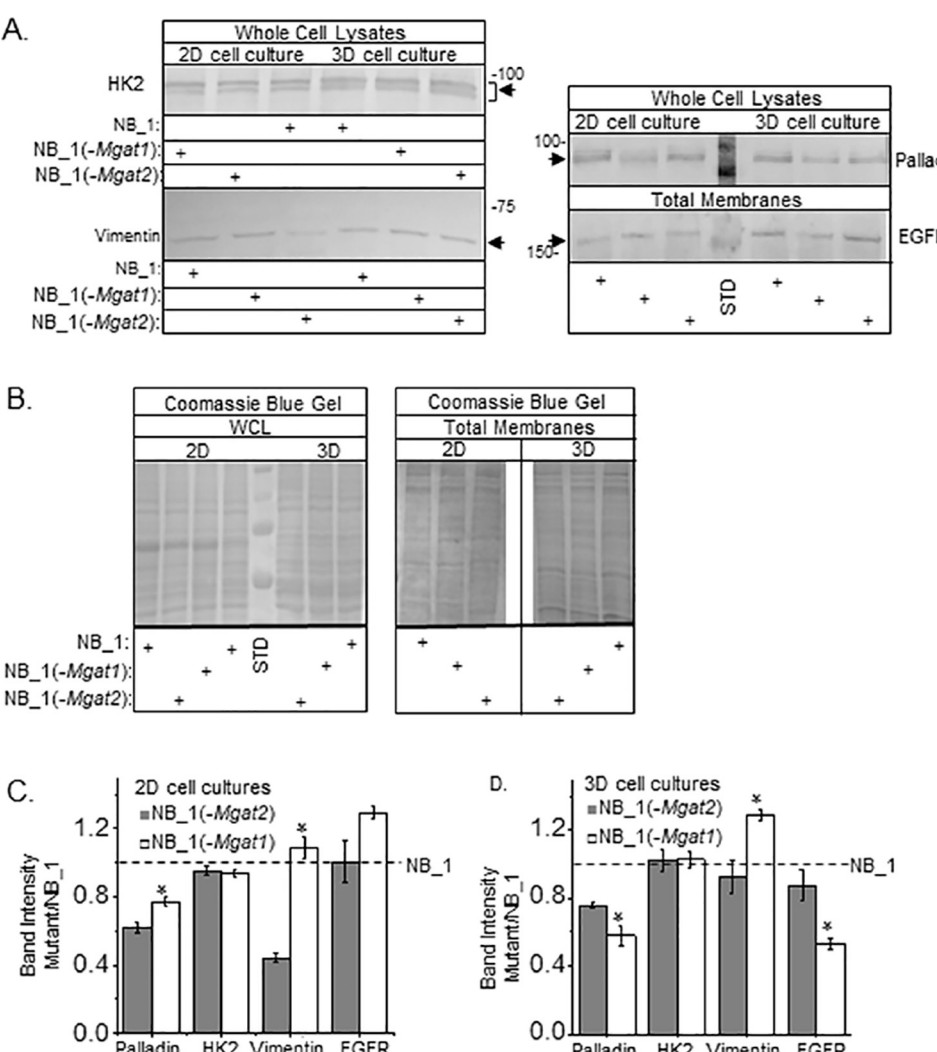

**Fig 9. Palladin, vimentin and EGFR levels were altered by changes in the N-glycosylation pathway.** (A) Western blots of HK2, palladin, and vimentin in whole cell lysates (WCL), and EGFR in total membranes (TM) from 2D and 3D cell cultures. Number adjacent to blot represents molecular weight marker in kDa. Arrows point to immunobands of interest. Protein levels loaded per well for detection of vimentin, EGFR and HK2 in both 2D and 3D samples were 20 μg per well while those for palladin were 20 μg and 10 μg for 2D and 3D samples, respectively. Blot of palladin has molecular weight markers of 75 and 100 kDa in the center lane while that of EGFR has the 150 kDa marker. Standards were not removed to show 2D and 3D samples were run on same gel and blots developed in a similar manner. (B) Coomassie blue stained gels of whole cell lysates and total membranes from 2D (20 μg) and 3D (20 μg) cell cultures. Molecular weight standards (STD) in kDa: 250; 150; 100; and 75 from top to bottom. Immunobands from NB_1 (-*Mgat2*) and NB_1(-*Mgat1*) cell lines were normalized to those from the NB_1 cell line from 2D (C) and 3D (D) cultures. Data are represented as the mean±SEM and were compared by student's T-test (*p < 0.05). *n* values were 3 where *n* represents a band.

cell morphology, growth and invasion appear to be changed under different growth conditions and by modifications in the N-glycosylation pathway.

## Discussion

This study showed that knockdown of the GnT-I in NB cells modified cell morphology, growth, adhesion, and invasion, along with expression patterns of MMPs, HK2, EGFR, vimentin, and palladin proteins in 2D cell cultures. When comparing these cellular processes and

protein levels of the parental cell line (NB_1) to cell lines with the N-glycosylation pathway interrupted (NB_1(-*Mgat1*) and NB_1(-*Mgat2*)) under 3D cell cultured conditions, they were quite different from those under 2D conditions. Previously, we demonstrated that NB_1 [19] and NB_1(-*Mgat2*) [19, 20] cells expressed significant levels of complex and hybrid types N-glycans, respectively. Here, lectin binding studies supported that NB_1(-*Mgat1*) cells predominantly expressed oligomannose type N-glycans. Further higher order N-glycans could be restored by transiently transfecting NB_1(-*Mgat1*) cells with mouse *Mgat1*. Taken together, different types of N-glycan populations favored different aberrant cellular properties and were dependent on whether the NB cells were cultured under 2D or 3D conditions. Emphasizing tumor microenvironment on cell growth and invasiveness.

Unexpectedly, cell invasiveness was different between 2D cultures of dispersed cells versus 3D cultures of spheroid cells, consisting of small, intermediate, and large size spheroids. Previously, we showed that the NB cell line with complex type N-glycans was more invasive than NB cells expressing more hybrid type N-glycans [18, 20]. Additionally, 2D cell culture studies demonstrated that complex N-glycans with β1,6-branches, along with a decline in hybrid type N-glycans, were detected in cells derived from high risk neuroblastoma (NLF) compared to cells derived from low risk neuroblastoma (SY5Y) [25]. Our current 2D cell culture study shared similarities to these results, and also, showed that NB cells with less processed type of N-glycans (oligomannose) were even less invasive. On the contrary, the NB cell line predominantly expressing oligomannose type N-glycans was more invasive than NB_1 and much more invasive than NB_1(-*Mgat2*) under 3D culture conditions. Sphere area measurements support that cell invasiveness was dependent on size. As expected, longer growth of the spheroids in matrigel increased cell invasiveness. Although spheroid size at 1, 4 and 6 days of culturing are relatively similar for NB_1 and NB_1(-*Mgat1*), the sphere area used for analysis was smaller for the latter. In general, the larger spheroids from the NB_1(-*Mgat1*) cell line were omitted since their protrusions from the spheroid would attach to the bottom of the plate causing attached cells to readily spread. Again, reinforcing the relative great cell invasiveness of NB_1 (-*Mgat1*) cells. Based on our 3D cell invasion studies, NB cells expressing complex N-glycans were less invasive than those expressing exclusively oligomannose. As such, our current results support a past clinical NB study which showed that GnT-V, the enzyme which initiates β1,6-branchpoints, was of higher abundance in favorable stages (1, 2 and 4s) than unfavorable stages (3 and 4) of NB [26]. Thus, it would be of great interest to evaluate whether unfavorable NB patients have lowered expression of GnT-I in tumor tissues compared to those with favorable NB. Further the cell culturing technique appears to have a major impact on interpreting cell invasiveness of NB cells.

A critical aspect of cell invasion is remodeling of the ECM which is carried out by the matrix metalloproteinase (MMP) family [27]. Under 2D cell culturing conditions, 3 gelatinase activities were identified when cells were cultured at least 72 h. Previously, we showed the lower band was MMP2 and was more abundant in the more invasive NB_1 cells than NB_1 (-*Mgat2*) cells when cultured for 48 h [18]. The current study verifies higher MMP activities in the more invasive cell line, NB_1, than NB_1(-*Mgat2*) at the later timepoint, however, the patterns of MMP activity were different. Additionally, the activity appeared lowest in the least invasive cell line, NB_1(-*Mgat1*). When 3D cell cultures were examined two of the three MMP activities observed in 2D cell cultures were identified and both activity bands were most intense in NB_1(-*Mgat1*) cells. These results supported higher expression of MMPs in the more invasive cell lines. Previously, we suggested that NB_1(-*Mgat2*) cells may have lower MMP levels due ER stress caused by a build-up of N-glycosylated proteins in the ER of 2D cultures [18] since anti-cancer compounds have correlated ER stress with lower MMP levels [28, 29]. However, our current study which reveals higher cell invasiveness and abundance of

MMP activities of NB_1(-*Mgat1*) cells under 3D culturing conditions opposes this explanation.

Cell morphology differs in 2D and 3D cell cultures. When the N-glycosylation pathway was interrupted in NB cells by silencing *Mgat1* (Fig 2A) or *Mgat2* [18], cells rounded up and neurites were truncated relative to NB cells with an intact N-glycosylation pathway in 2D cultures. Expression levels of two cytoskeleton proteins, vimentin and palladin, were also altered in the N-glycosylation mutant cells relative to parental cells. Past studies have shown the roles of vimentin [30, 31] and palladin [13] in cell shape, and furthermore in lengthening of neurites of neurons. As such, the truncated neurites are likely due to reduced levels of palladin in 2D cell cultures of NB_1(-*Mgat1*) and both palladin and vimentin of NB_1(-*Mgat2*). It appears palladin has a stronger influence on neurite formation as vimentin levels in NB_1(-*Mgat1*) cells were similar to NB_1 cells. In 3D cultures, palladin and vimentin levels were reduced in NB_1 (-*Mgat2*) while palladin levels declined and vimentin levels were raised in NB_1(-*Mgat1*) relative to those in NB_1. A notable difference in the morphology of cells in the outer region of spheroids was their diameter. The size of NB_1, NB_1(-*Mgat2*) and NB_1(-*Mgat1*) cells was largest, intermediate, and smallest, respectively. Interestingly, the size of cells at the edge of the spheroid correlated with the levels of palladin as this protein was lower in NB_1(-*Mgat2*) cells and lowest in NB_1(-*Mgat1*) cells relative to NB_1 cells. In comparing 2D versus 3D cultures, the level of palladin levels was higher in 3D. Taken together, our results support the role of palladin in neurite lengthening of 2D cultures, and furthermore supports novel roles of palladin in size of cells in the outer portion of the spheroid cells, and production of larger spheroid cells.

Cell growth occurred at a faster rate in NB cells with complex type N-glycans relative those without complex type N-glycans but this process was retarded in 3D cultures compared to 2D cultures. NB_ 1 and NB_1(-*Mgat2*) cell lines showed about a 2-fold difference in quantity of cells after 2 and 13 days under 2D and 3D conditions, respectively [18, 20]. The decrease in cell growth has been previously described for other cell lines [8]. Here a similar trend was shown in that a 2-fold difference between NB_1 and NB_1(-*Mgat1*) cells occurred in around 1 day in monolayer cell cultures while this difference took roughly 13 and 9 days in spheroid cell formation in agar versus in 5D spherical plates, respectively. Previously, cell growth of NB_ 1 and NB_1(-*Mgat2*) cell lines were similar to the cell proliferation rates in 2D cultures [20]. 2D cell culture involves cell-attached growth with cells having relatively equal exposure to media, while 3D cultured cells comprise of non-adhered cell aggregates with cells in the outer spheroid shell verses those in the inner core having maximal to minimal exposure to media [8]. Here, 5D sphericalplates were used for 3D cultures which have been shown to generate spheroids, lacking necrotic cores [32]. Additionally, protein expression of 3D versus 2D NB cultures are quite different, indicating the requirement of different cellular processes for their replication and viability [5]. Here the amount of HK2 levels was more abundant in 3D than 2D cell cultures which support the increased levels of other glycolytic enzymes, such as pyruvate kinase α-enolase, phosphoglycerate mutase 1 and triphosphate isomerase [5]. Thus, the difference in cell growth and the additional breakdown of glucose for energy between 2D and 3D cell cultures are likely supported by their growth requirements. Under 2D culturing, cell proliferation is the major factor affecting NB_1 cell growth while other processes are involved in the rate of cell growth under 3D conditions, such as ECM remodeling, transport, and cell adhesion.

EGFR, a receptor protein involved in cell proliferation and invasiveness, is known to be overexpressed in human neuroblastoma cells [16, 17]. Herein, we observed a decrease in cell proliferation when N-glycans were less processed, although EGFR levels remain unchanged under 2D culturing conditions. We suggest that the slowed cell proliferation could be due to

truncated N-glycans as enhanced branching of the N-glycans attached to EGFR increased cell growth and proliferation under 2D conditions [33]. In 3D cell cultures, significantly lower levels of EGFR were detected in the N-glycosylation mutant cell line NB_1(-*Mgat1*) relative to NB_1 and NB_1(*Mgat2*) cell lines. However, spheroid cell growth between NB_1(*Mgat2*) and NB_1(*Mgat1*) was somewhat similar, suggesting additional processes and factors are involved in controlling cell growth.

We conclude, although the variables involved in neuroblastoma growth and progression are still poorly understood, engineered N-glycosylation mutant neuroblastoma (NB) cell lines provide a fundamental knowledge base as to the roles of various N-glycan types and their influence on NB development and prognosis, as well as potentially aid in development of additional treatment options. Based on our current 2D and 3D results of parental and N-glycosylation mutant NB cell lines, there are considerable differences in cellular processes, with emphasis on cell invasiveness and growth, and furthermore expression levels of cell growth proteins, cytoskeleton proteins and ECM digesting enzymes which likely support the value of 3D cell cultures in cancer research as they more closely mirror tumors in tissues.

## Supporting information

**S1 Raw images.**
(PDF)

**S1 Data. Group data of Figs 1–9.**
(XLSX)

## Author Contributions

**Conceptualization:** M. Kristen Hall, Ruth A. Schwalbe.

**Data curation:** M. Kristen Hall, Adam P. Burch, Ruth A. Schwalbe.

**Formal analysis:** M. Kristen Hall, Adam P. Burch, Ruth A. Schwalbe.

**Funding acquisition:** Ruth A. Schwalbe.

**Investigation:** M. Kristen Hall, Adam P. Burch, Ruth A. Schwalbe.

**Methodology:** M. Kristen Hall, Adam P. Burch, Ruth A. Schwalbe.

**Project administration:** Ruth A. Schwalbe.

**Resources:** Ruth A. Schwalbe.

**Software:** Ruth A. Schwalbe.

**Supervision:** Ruth A. Schwalbe.

**Validation:** M. Kristen Hall, Ruth A. Schwalbe.

**Visualization:** M. Kristen Hall, Ruth A. Schwalbe.

**Writing – original draft:** M. Kristen Hall, Adam P. Burch, Ruth A. Schwalbe.

**Writing – review & editing:** M. Kristen Hall, Adam P. Burch, Ruth A. Schwalbe.

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
