## [Decision Letter · Decision Letter 0]

18 Sep 2021

PONE-D-21-18235Functional analysis of N-acetylglucosaminyltransferase-I knockdown in 2D and 3D neuroblastoma cell culturesPLOS ONE

Dear Dr. Schwalbe,

Thank you for submitting your manuscript to PLOS ONE. After careful consideration, we feel that it has merit but does not fully meet PLOS ONE’s publication criteria as it currently stands. Therefore, we invite you to submit a revised version of the manuscript that addresses the points raised during the review process. Please note the extensive comments from Reviewer No. 1.  If these issues can be addressed, it is likely that this manuscript will be accepted for publication.

We look forward to receiving your revised manuscript.

Kind regards,

Salvatore V Pizzo

Academic Editor

PLOS ONE

Journal Requirements:

**3. **PLOS ONE now requires that authors provide the original uncropped and unadjusted images underlying all blot or gel results reported in a submission’s figures or Supporting Information files. This policy and the journal’s other requirements for blot/gel reporting and figure preparation are described in detail at https://journals.plos.org/plosone/s/figures#loc-blot-and-gel-reporting-requirements and https://journals.plos.org/plosone/s/figures#loc-preparing-figures-from-image-files. When you submit your revised manuscript, please ensure that your figures adhere fully to these guidelines and provide the original underlying images for all blot or gel data reported in your submission. See the following link for instructions on providing the original image data: https://journals.plos.org/plosone/s/figures#loc-original-images-for-blots-and-gels. In your cover letter, please note whether your blot/gel image data are in Supporting Information or posted at a public data repository, provide the repository URL if relevant, and provide specific details as to which raw blot/gel images, if any, are not available. Email us at plosone@plos.org if you have any questions.

Reviewers' comments:

Reviewer's Responses to Questions

**Comments to the Author**

1. Is the manuscript technically sound, and do the data support the conclusions?

Reviewer #1: Yes

2. Has the statistical analysis been performed appropriately and rigorously? 

Reviewer #1: Yes

3. Have the authors made all data underlying the findings in their manuscript fully available?

Reviewer #1: Yes

4. Is the manuscript presented in an intelligible fashion and written in standard English?

Reviewer #1: Yes

5. Review Comments to the Author

Reviewer #1: In the manuscript by Hall et al., together with their previous paper (ref. 18-20) the authors generated Mgat1 (encodes GnT-I) knock out neuroblastoma cells by CRIPR/Cas9 and obtained mutant cells with aberrant N-glycosylation. The mutant cells showed altered cell morphology and growth properties compared to wild-type parental cells. The authors also showed that the expression of several proteins including MMP, receptor-tyrosine kinase and cytoskeletal proteins were significantly changed in mutant cells.

The experiment seems to be well designed and conducted, and the results ae convincing to support their conclusion. The manuscript is well constructed and written in precise English.

Even though the manuscript largely lacks the mechanistic insights how N-glycans regulate and what proteins is most responsible for these cellular phenotypes, the phenomena itself might be informative to the reader of glycobiology and tumor biology.

I think the publication of the manuscript can be considered if the following concerns are addressed.

Major point

1. through the manuscript

GnT-1 and GnT-2 should be GnT-I and GnT-II (Roman, not Arabian)

2. The schematic drawing to show N-glycans structures and how GnT-I and GnT-II work on the structures should be briefly presented on Figure 1A for beginner readers.

3. Figure 1D

The samples should be run on the same gel and blotted on the same membrane for each blot.

4. Figure 9A

The image is very poor and the samples should be run on the same gel and blotted on the same membrane for each blot.

5. The authors should more discuss the relevance of their finding to the human clinical situations. For instances, human neuroblastoma is classified into stage 1 to 4 and 4S. Do you have any information about the expression of GnT-I with different stages?

Minor point

Lime 108

paladin>paladin

Line 132

two dimensional (2D) versus three dimensional (3D) > 2D and 3D

the abbreviations were already defined and not needed here

Figure 1C

“NB_1 (Mgat1)” should be “NB_1 (-Mgat1)”

Figure 2A

the scale bar should be included into the image

Figure 2B

What is OG? Outgrowth? “Neurite Length” and “Neurite Width” are better.

Figure 2D

Need the scale bar.

Figure 3B

Need the scale bar.

Figure 3C Y-axis

What is the unit?

Figure 3D

Need the scale bar.

Figure 5A and 5B

Need the scale bar

Figure 5C, Y-axis

Should be “Cell invasion index (ratio to spheroid size)”

Figure 5D, Y-axis

Need the unit pixels?

Figure 6

See the comment on Figure 5.

Figure 7B and 7D, Y-axis

Should be “Gelatinase activity (Mutant/NB_1)

Figure 8A

The cell line names are missing.

Figure 9C and 9D, Y-axis

Should be “Band intensity (Mutant/NB_1)” or “Relative expression (Mutant/NB_1)”.

6. PLOS authors have the option to publish the peer review history of their article (what does this mean?). If published, this will include your full peer review and any attached files.

Reviewer #1: **Yes: **Kazuma Sakamoto

---

## [Author Response · Author response to Decision Letter 0]

14 Oct 2021

Point-by-point comments to reviewer

5. Review Comments to the Author

We wish to thank the reviewer for a thorough review as the comments have improved the manuscript.

Authors’ answers to the questions above. 

The answers to questions made by reviewer were positive. However, we have addressed availability of group data supporting the mean. Groups data has been uploaded as supplementary group data excel file. Also, we have attached a file: S1_Raw_images which has unmodified images of western blot, lectin blots, gelatinase gels and coomassie blue stained SDS gels.

Reviewer #1: In the manuscript by Hall et al., together with their previous paper (ref. 18-20) the authors generated Mgat1 (encodes GnT-I) knock out neuroblastoma cells by CRIPR/Cas9 and obtained mutant cells with aberrant N-glycosylation. The mutant cells showed altered cell morphology and growth properties compared to wild-type parental cells. The authors also showed that the expression of several proteins including MMP, receptor-tyrosine kinase and cytoskeletal proteins were significantly changed in mutant cells.

The experiment seems to be well designed and conducted, and the results ae convincing to support their conclusion. The manuscript is well constructed and written in precise English.

Even though the manuscript largely lacks the mechanistic insights how N-glycans regulate and what proteins is most responsible for these cellular phenotypes, the phenomena itself might be informative to the reader of glycobiology and tumor biology.

I think the publication of the manuscript can be considered if the following concerns are addressed.

Major point

1. through the manuscript

GnT-1 and GnT-2 should be GnT-I and GnT-II (Roman, not Arabian)

Authors response: This has been changed throughout the manuscript.

2. The schematic drawing to show N-glycans structures and how GnT-I and GnT-II work on the structures should be briefly presented on Figure 1A for beginner readers.

Authors response: A schematic drawing has been added as Figure 1A. The figure legend was updated, lines 300-303. The figure is stated in the text of the results, lines 292-294.

3. Figure 1D

The samples should be run on the same gel and blotted on the same membrane for each blot.

Figure 1D was changed to Figure 1E due to insertion of the schematic drawing requested by reviewer. The samples were run on the same blot as shown and can also be viewed in the S1_Raw_images file. We have removed lines on blots for clarification.

4. Figure 9A

The image is very poor and the samples should be run on the same gel and blotted on the same membrane for each blot.

The samples were run on the same blot as shown and can also be viewed in the S1_Raw_images file. For clarification purposes, we have removed lines on blots. In some cases, the 2D and 3D samples were separated by standards as shown in figure. The standard was included on blot to illustrate the samples were on the same blot.

5. The authors should more discuss the relevance of their finding to the human clinical situations. For instances, human neuroblastoma is classified into stage 1 to 4 and 4S. Do you have any information about the expression of GnT-I with different stages?

Currently, information is lacking about the expression of GnT- I with the various stages of NB. However, we have elaborated on GnT-I in favorable and unfavorable NB in the 2nd paragraph of the discussion.

“… Again, reinforcing the relative great cell invasiveness of NB_1(-Mgat1) cells. Based on our 3D cell invasion studies, NB cells expressing complex N-glycans were less invasive than those expressing exclusively oligomannose. As such, our current results support a past clinical NB study which showed that GnT-V, the enzyme which initiates β1,6-branchpoints, was of higher abundance in favorable stages (1, 2 and 4s) than unfavorable stages (3 and 4) of NB [26]. Thus, it would be of great interest to evaluate whether unfavorable NB patients have lowered expression of GnT-I in tumor tissues compared to those with favorable NB. Further the cell culturing technique appears to have a major impact on interpreting cell invasiveness of NB cells.”

Minor point

Lime 108

paladin>paladin 

This has been corrected.

Line 132

two dimensional (2D) versus three dimensional (3D) > 2D and 3D

the abbreviations were already defined and not needed here

This has been corrected.

Figure 1C

“NB_1 (Mgat1)” should be “NB_1 (-Mgat1)”

This has been corrected.

Figure 2A

the scale bar should be included into the image

A scale bar (in µm) has been added. 

Figure 2B

What is OG? Outgrowth? “Neurite Length” and “Neurite Width” are better.

Outgrowth has been changed to neurite. Also pixels have been converted to µm.

Figure 2D

Need the scale bar.

A scale bar (in µm) has been added. 

Figure 3B

Need the scale bar.

A scale bar (in µm) has been added. 

Figure 3C Y-axis

What is the unit? 

µm2

Figure 3D

Need the scale bar.

A scale bar (in µm) has been added. 

Figure 5A and 5B

Need the scale bar

A scale bar (in µm) has been added. 

Figure 5C, Y-axis

Should be “Cell invasion index (ratio to spheroid size)”

Authors reply: The y-axis represents invasion area/sphere area.

Since there appeared to be some confusion, we changed the figure panels. Figure 5B shows how the invasion was calculated (invasion=invasive area – sphere area). The y-axis of figure 5C was changed to invasion area/sphere area. A reference for the analysis has been added in the methods (Berens EB, Holy JM, Riegel AT, Wellstein A. A Cancer Cell Spheroid Assay to Assess Invasion in a 3D Setting. Journal of visualized experiments : JoVE. 2015;(105). Epub 2015/12/10. doi: 10.3791/53409. PubMed PMID: 26649463; PubMed Central PMCID: PMCPMC4692745) (see line 244). Panel D represent plots of sphere area versus invasion area for each cell line to determine Pearson’s correlation coefficients between these two variables. The linear fits show a positive and significant correlation of the two variables, and the rationale for representing cell invasiveness of the different cell lines using the invasion/sphere versus cell line plot. The figure legend and results section has been changed accordingly.

Figure 5D, Y-axis

Need the unit pixels?

This panel has been removed and sphere area versus invasion area plot has been added. The units are in µm. 

Figure 6

See the comment on Figure 5.

Scale bar has been added. The y-axis of Panels D,E,F have been changed as described above for figure 5. Panels with bar graphs of sphere area versus cell line have been removed. 

Figure 7B and 7D, Y-axis

Should be “Gelatinase activity (Mutant/NB_1)

The label of Y-axis has been changed.

Figure 8A

The cell line names are missing.

The names of cell lines have been added.

Figure 9C and 9D, Y-axis

Should be “Band intensity (Mutant/NB_1)” or “Relative expression (Mutant/NB_1)”.

The y-axis was changed to Band intensity (Mutant/NB_1).

---

## [Decision Letter · Decision Letter 1]

26 Oct 2021

Functional analysis of N-acetylglucosaminyltransferase-I knockdown in 2D and 3D neuroblastoma cell cultures

PONE-D-21-18235R1

Dear Dr. Schwalbe,

We’re pleased to inform you that your manuscript has been judged scientifically suitable for publication and will be formally accepted for publication once it meets all outstanding technical requirements.

Kind regards,

Salvatore V Pizzo

Academic Editor

PLOS ONE

Additional Editor Comments (optional):

Reviewers' comments:

Reviewer's Responses to Questions

**Comments to the Author**

1. If the authors have adequately addressed your comments raised in a previous round of review and you feel that this manuscript is now acceptable for publication, you may indicate that here to bypass the “Comments to the Author” section, enter your conflict of interest statement in the “Confidential to Editor” section, and submit your "Accept" recommendation.

Reviewer #1: All comments have been addressed

2. Is the manuscript technically sound, and do the data support the conclusions?

Reviewer #1: Yes

3. Has the statistical analysis been performed appropriately and rigorously? 

Reviewer #1: Yes

4. Have the authors made all data underlying the findings in their manuscript fully available?

Reviewer #1: Yes

5. Is the manuscript presented in an intelligible fashion and written in standard English?

Reviewer #1: Yes

6. Review Comments to the Author

Reviewer #1: The authors well addressed my concerns.

I do not have additional requirements and recommend the publication of the manuscript in PLOS One.

7. PLOS authors have the option to publish the peer review history of their article (what does this mean?). If published, this will include your full peer review and any attached files.

Reviewer #1: **Yes: **Kazuma Sakamoto

---

## [Editor Report · Acceptance letter]

28 Oct 2021

PONE-D-21-18235R1 

Functional analysis of N-acetylglucosaminyltransferase-I knockdown in 2D and 3D neuroblastoma cell cultures 

Dear Dr. Schwalbe:

I'm pleased to inform you that your manuscript has been deemed suitable for publication in PLOS ONE. Congratulations! Your manuscript is now with our production department. 

Kind regards, 

on behalf of

Dr. Salvatore V Pizzo 

Academic Editor

PLOS ONE